# Do medical students and residents impact the quality of patient care? An assessment from different stakeholders in an Italian academic hospital, 2019

Giuseppe Perri[1][○][¤], Matteo d'Angelo[1][○], Cecilia Smaniotto[1][○]*, Massimo Del Pin[1][○], Edoardo Ruscio[1][○], Carla Londero[2][○], Laura Brunelli[1,2][○], Luigi Castriotta[3][○], Silvio Brusaferro[1][○]

1 Department of Medicine, University of Udine, Udine, Italy, 2 Accreditation, Clinical Risk Management and Performance Assessment Unit, Friuli Centrale Healthcare University Trust, Udine, Italy, 3 Institute of Hygiene and Clinical Epidemiology, Friuli Centrale Healthcare University Trust, Udine, Italy

○ These authors contributed equally to this work.
¤ Current address: Hospital Management, Giuliano Isontina Healthcare University Trust, Trieste, Italy
* smaniotto.cecilia@spes.uniud.it

**Data Availability Statement:** All relevant data are within the manuscript and its Supporting Information files.

## Abstract

Medical students and residents play an important role in patient care and ward activities, thus they should follow hospital procedures and ensure best practices and patient safety. A survey concerning staff on training was conducted to assess the perceived quality of healthcare from healthcare workers (HCWs), residents, medical students and patients in Udine Academic Hospital, Italy. Between December, 2018 and March, 2019, a 5-point Likert-scale questionnaire was administered in 21 units, covering four thematic areas: patients and medical staff satisfaction with the quality of care provided by residents and students, patient privacy, clinical risk management, patient perception of staff on training. Data analysis included descriptive analysis and ordered logistic regressions. A total of 596/1,863 questionnaires were collected from: HCWs (165/772), residents (110/355), students (121/389), and patients (200/347). Residents were rated high both by patients (median = 5, IQR = 4–5, OR 0.49, 95%CI 0.26–0.93) and HCWs (median = 4, IQR = 3–5, OR 0.14, 95%CI 0.08–0.26), with a lower score for medical students on the same topic, both by patients (median = 4, IQR = 3–5, OR 2.94, 95%CI 1.49–5.78) and HCWs (median = 3, IQR = 2–3, OR 0.41, 95%CI 0.25–0.67). Therefore, the role of staff on training in quality and safety of healthcare deserves integrated regular evaluation, since direct interaction with patients contributes to patients' perception of healthcare.

## Introduction

Medical studies require an effective planning to properly manage such a complex didactic activity. Therefore, it is important to ensure a collaborative, constructive, and respectful link

**Funding:** The authors received no specific funding for this work.

**Competing interests:** The authors have declared that no competing interests exist.

between academic activities and healthcare needs and tasks in providing care; it is equally important to design a formative assessment to evaluate the overall effectiveness of the programme and its elements.

Concerning the undergraduate level, basic medical training in Italian University takes place mainly in the last three years of academic education (out of six). After graduation, physicians attend a four- or five-year residency programme, in which each specialty establishes a specific curriculum and training activities to meet national requirements. The Italian reform of residency training programmes, established by D.L.vo 257/91, which implements European Directive 82/76/EEC, requires full-time participation of medical residents in clinical practice within the healthcare facility where they are attending their residency programme. The aim of this law is to provide residents with comprehensive and relevant training so that they achieve an appropriate level of competence and performance. Moreover, the D.L.vo 502/92 regulates the relationship between the National Health System (NHS) and Italian University, stating that each Region shall establish specific agreements with Universities to regulate the contribution of Faculties of Medicine and Surgery to NHS healthcare activity, fully respecting their scientific and didactic aims as an institution. Therefore, both academic and non-academic physicians must fulfil duties related to healthcare, didactic and research activities [1]. Medical students and residents (described as "staff on training" from this point forward when referring to both categories together), during their training, interface both with patients and HCWs and come into contact with different multidisciplinary work teams and contexts. Adherence to safety standards and best clinical practices observed by the hospital, as well as awareness of the risks associated with hands-on activities, are essential elements for their stay on the wards and departments. To ensure quality and safety of healthcare services, while guaranteeing valid clinical training paths, it is essential to understand how the behaviour and actions of staff on training are perceived by both patients and HCWs in relation to safety and clinical risk management.

According to patient-as-partner approach, after receiving the necessary information to choose the degree of control over health decisions that affect them, patients are directly involved in improving the quality in healthcare services [2]. Moreover, the system should be able to take into account the different preferences of patients and encourage shared decision making [3, 4].

The importance of the assessment of perceived quality by patients within healthcare facilities has increased since the 80s, along with transparency and accountability in morbidity and mortality data [5]. This concept has been recently widened to include an improvement in relationships between patients, physicians and caregivers to ensure consideration of stakeholders' preferences, needs and values [6]. This reflects the increasing trend of using data on satisfaction and perceived quality to define medical outcomes, as well as the growing relation between public perceptions of quality and expert ratings medical effectiveness [5].

The patient is thus an active member of the healthcare system, in accordance with the assumption that the subject is a co-producer (prosumer) of health [7] and a protagonist of his/her own healthcare path. Since patients' satisfaction is considered an outcome of healthcare [8], the perspective of user-prosumer becomes a measurable indicator whose value can be used to guide healthcare decision makers. However, as the impact and degree of success of such strategies are also influenced by the different contexts in which the interventions are planned and implemented [9], aspects such as leadership, appreciation of personal skills and organisational resources cannot be overlooked. Moreover, such assessment allows both the application of quality principles and the achievement of users' satisfaction [10], although this cannot be sufficient, since the user is not necessarily an expert [11]. In general, users express particular agreement on those interventions perceived as directly improving their own health, such as

solving a problem, relieving painful symptoms, improving an impaired performance status, etc., but pay equal attention to several other aspects of the service [10, 12], including: timing, clarity of procedures, information about the treatment, orientation and reception in the facility, features of the facility, social and human relations.

Even though the positive feedback from patients to participation in medical training is widely reported in the literature [13], according to Barksby [14] it is necessary to consider the impact of training staff presence on patients, also to optimise resources for training future medical professionals. On the other hand, the training staff's opinions collected concerning actions and compliance with safety standards set by the hospital can be usefully compared with the perception of patients and HCWs. This type of patient-centred survey can provide suggestions for leadership and provide a contribution to adopt coherent strategies for healthcare quality and safety and training effectiveness, as well as satisfaction of all stakeholders [15].

The main aim of this study was to evaluate the quality of care in an academic hospital, taking into account different perspectives from: staff on training, patients and HCWs; secondary objectives were to identify elements that contribute to the quality and safety of healthcare, as well as strategies to optimise the educational goals in healthcare.

## Methods

### Design of the study and previous research

This monocentric, observational and cross-sectional study was developed by the Quality Team from Udine Academic Hospital and is based on a pilot project conducted in 2017 with 279 contributions.

### Design of the questionnaire and testing phase

A multiple-choice questionnaire was constructed based on the pilot project and available literature for each category of stakeholders: medical students, medical residents, HCWs and patients [16]. The questions covered four thematic areas: patient and medical staff satisfaction with the quality of care provided by residents (A); patients' privacy (B); clinical risk management (C); patients' perceptions of staff on training (D). The number of questions for each macro area varied depending on the category of participants: a total of 30, 46, 24, and 31 questions were asked to patients, HCWs, medical students and residents, respectively. Demographic information such as age, gender, level of education (patients), hospital unit (patients, HCWs), year of attendance (staff on training) and academic tutoring (HCWs) were collected. Responses were coded using a Likert scale score expressed as level of agreement: strongly agree (5), agree (4), neither agree nor disagree (3), disagree (2), strongly disagree (1). The comprehensibility and internal consistency of the questionnaire were tested by presenting it to a group of multidisciplinary experts, working for the Quality team (Cronbach alpha = 0.85). During this testing phase, the time to complete the questionnaire was set at 30 minutes, taking into account the heterogeneous psychophysical conditions of the respondents.

### Inclusion and exclusion criteria

Patients, HCWs and staff on training of the following units of Udine Academic Hospital participated in the survey: general surgery, internal medicine, obstetrics and gynaecology, oral and maxillofacial surgery, plastic surgery, ophthalmology, cardiology, pneumology, nephrology, otolaryngology, orthopaedics, urology, infectious diseases, rheumatology, vascular surgery, gastroenterology, haematology, neurosurgery, cardiac surgery, thoracic surgery, accident and emergency departments. To participate in the survey, inpatients had to be ≥18 years old,

admitted to Udine Academic Hospital at least since the night before the index day, able to respond, and willing to participate in the survey. Patients who were not available for other examinations or procedures were excluded. All medical residents attending Udine Academic Hospital were included. As for medical students, only those attending their 4th-5th-6th year at University of Udine were included, since during this period they were carrying out their clinical training in the hospital facilities and thus they were in contact with patients. All participants were informed about the aims of the study and the confidentiality of data. The questionnaires of medical students, residents and HCWs' were completely anonymous and participation in the study was voluntary. Participating patients signed an informed consent form before completing the anonymous questionnaire. The study protocol was approved by Friuli-Venezia Giulia Regional Unique Ethical Committee (CEUR, D 2596 18/12/2018).

## Data collection

HCWs, medical students and medical residents were enrolled between January and March, 2019 by sending an email invitation to participate with a direct link to the online questionnaire. The study was also announced by posters display with QR code/link to the questionnaire on notice boards in the hospital. Questionnaires were administered to inpatients between December, 2018 and March, 2019. Ten trained clinicians distributed the questionnaires to inpatients and were available to provide any necessary explanation until the survey was completed.

## Sample size calculation

Convenience sampling was chosen for data collection. The minimum number of needed questionnaires was calculated for each respondent category using the OpenEpi tool, assuming a precision of 10%, 95% confidence interval, expected response rate to the first pivotal question in the questionnaire and reference population sizes; for each respondent category were:

- 76 questionnaires from inpatients (population size: 350; expected response rate 50%);

- 74 questionnaires from HCWs (population size: 800; expected response rate 70%);

- 68 questionnaires from medical students (population size 400; expected response rate 70%);

- 68 questionnaires from medical residents (population size 400, expected response rate 70%).

## Data analysis

All responses were entered into an electronic spreadsheet programme and checked for any incorrect information or missing values. Completed questionnaires were aggregated according to stakeholder categories, medical, surgical or diagnostic and clinical services department. The study population features were investigated performing descriptive statistics on categorical and numerical variables. Frequency distributions were used for categorical variables. For numerical variables we considered mean, median, interquartile range (IQR), standard deviation (SD). Ordered logistic regression were performed to investigate the association between answers and category, after assessing proportionality odds assumption through the approximate likelihood-ratio test of proportionality of odds; when the assumption did not hold generalised ordered logistic regression were performed (using "autofit" option when questions were answered by 3 or more categories) [17]. Scores of two question were recoded as dichotomous "Agree" (strongly agree (5), agree (4)) and "Don't agree" (neither agree nor disagree (3), disagree (2), strongly disagree (1)) for lack of convergence; logistic regressions' goodness of fit

Table 1. Age of respondents, according to respondent category.

| Respondents | | Mean Age [std. dev] (years) | | |
|---|---|---|---|---|
| | | Male n = 232 | Female n = 347 | Total n = 596 |
| Patients n = 347 | | 63.5 [16.2] | 59.5 [18.1] | 61.4 [17.3] |
| Staff on training n = 744 | Medical students n = 389 | 24.4 [3.1] | 24.3 [1.9] | 24.3 [2.5] |
| | Medical residents n = 355 | 30.2 [3.7] | 29.2 [2.2] | 29.6 [2.9] |
| Healthcare workers n = 772 | Physicians n = 295 | 48.7 [10.2] | 42.1[7.1] | 45.9 [9.5] |
| | Nurses n = 477 | 37 [7.6] | 38.3 [9.1] | 38.2 [8.9] |

was measured with Hosmer-Lemeshow test. Data analysis was performed using Stata 17 (Stata-Corp. 2021. Stata Statistical Software: Release 17. College Station, TX: StataCorp LLC).

## Results

Participation in the study was offered to 1,863 subjects, including 389 medical students, 355 medical residents, 772 HCWs (295 physicians and 477 nurses), and 347 patients. In all, 596 questionnaires were collected: 121 from medical students (21%), 110 from medical residents (19%); 165 from HCWs (28.5%), of whom 44 (27%) were physicians and 121 (73%) were nurses, and 200 from patients (35%). The overall response rate was 32.1%. Table 1 (below) reports mean age of all respondents for each group.

More than half of the patients (51%, 100/196) had ISCED (International Standard Classification of Education) level >3; specifically, 23% (45/196) had primary education, 26% (51/196) had lower secondary education and 37% (73/196) had upper secondary education, while 14% (27/196) had tertiary or equivalent education. The respondents among inpatients were 44% (88/200) from medical department and 56% (112/200) from surgical department. A similar distribution was observed for HCWs category, as 51% of them worked in the medical department (85/165). Residents working within medical department were 55% (60/110), while 20% (22/110) of them were from surgical and 25% (27/110) from diagnostic and clinical services departments. Twenty-six medical students (21%) were attending the 4[th], 29 (24%) the 5[th] and 66 (55%) the 6[th] year of their medical studies.

### Patients and medical staff satisfaction with the quality of care provided by residents and medical students

Residents' self-perception of their contribution to the quality of healthcare (median[IQR] = 5 [5–5]) is significantly higher than the assessment given by patients (median[IQR] = 5[4–5]; OR[95% CI] = 0.49 [0.26–0.93]) and HCWs (4[3–5]; OR[95%CI] = 0.14 [0.08–0.26]) (Table 2, Question A1). Instead, patients rate the contribution of students higher (4[3–5]; OR[95%CI] = 2.94 [1.49–5.78]) than medical students themselves (3[3–4]), but also compared to residents and HCWs.

The HCWs rate students' ability to influence the human side of care as adequate (3[3–4]), but are more certain (5[4–5]) about the residents' contribution on this point (Table 2, Question A2).

### Patient privacy

There is agreement that residents pay close attention to preserve patient privacy during the course of their clinical duties (Table 2, Question B1) and to avoid inadvertent disclosure of sensitive information (Table 2, Question B2), with patients scoring highest (Question B1: 5 [5–5]; OR[95%CI] = 3.17 [1.54–6.57]; Question B2: 5[5–5]; OR[95%CI] = 1.74 [0.56–5.40]) of all stakeholders on this issue.

**Table 2. Perception of staff on training by students, residents, patients and HCWs on questions posed, reported by single question as medians and IQR, as well as OR (95%CI).**

| Area | Question (perception on…) | Staff on training category | Perceived by… | | | | | | | |
| | | | Students (N = 389) | | Residents (N = 355) | | Patients (N = 347) | | HCWs (N = 722) | |
| | | | Median (IQR) | OR (95% CI) | Median (IQR) | OR (95% CI) | Median (IQR) | OR (95% CI) | Median (IQR) | OR (95% CI) |
| A | A1) Assistance improved by staff on training | Residents | 5 (5–5) | 0.78 (0.41–1.48) | 5 (5–5) | 1 | 5 (4–5) | 0.49 (0.26–0.93) | 4 (3–5) | 0.14 (0.08–0.26) |
| | | Students | 3 (3–4) | 1 | 3 (2–4) | 0.68 (0.43–1.09) | 4 (3–5) | 2.94* (1.49–5.78) | 3 (2–3) | 0.41 (0.25–0.67) |
| | A2) Human side assistance contribution | Residents | / | / | 4 (3–5) | 1 | / | / | 5 (4–5) | 1.54 (0.79–2.98) |
| | | Students | 4 (3–4) | 1 | / | / | / | / | 3 (3–4) | 0.42 (0.25–0.69) |
| B | B1) Preservation of privacy | Residents | / | / | 5 (4–5) | 1 | 5 (5–5) | 3.17 (1.54–6.57) | 5 (4–5) | 0.65 (0.39–1.10) |
| | | Students | 5 (4–5) | 1 | / | / | 5 (5–5) | 2.40 (1.03–5.61) | 4 (3–5) | 0.20 (0.12–0.34) |
| | B2) No accidental data diffusion | Residents | / | / | 5 (4–5) | 1 | 5 (5–5) | 1.74** (0.56–5.40) | 5 (4–5) | 0.36** (0.16–0.80) |
| | | Students | 5 (4–5) | 1 | / | / | 5 (4.5–5) | 3.19 (1.50–6.79) | 4 (3–5) | 0.39 (0.23–0.65) |
| C | C1) Medical record management | Residents | 5 (5–5) | 1 | 5 (4–5) | 1 | / | / | 4 (2–5) | 0.09* (0.03–0.22) |
| | | Students | / | / | / | / | / | / | 4 (3–5) | 0.09 (0.48–0.17) |
| | C2) Informed consent collection | Residents | 5 (5–5) | 1 | 5 (5–5) | 1 | / | / | 5 (4–5) | 0.04** (0.01–0.29) |
| | | Students | / | / | / | / | / | / | 4 (3–5) | 0.28* (0.13–0.60) |
| | C3) Reporting mistakes | Residents | / | / | 4 (4–5) | 1 | / | / | 3 (2–4) | 0.15 (0.08–0.27) |
| | | Students | 5 (4–5) | 1 | / | / | / | / | 4 (4–5) | 0.51 (0.32–0.82) |
| | C4) Hand washing | Residents | / | / | 4 (4–5) | 1 | 5 (4–5) | 1.39* (0.61–3.15) | 4 (3–4) | 0.16 (0.09–0.28) |
| | | Students | 4 (4–5) | 1 | / | / | 5 (5–5) | 5.12 (2.28–11.51) | / | / |
| | C5) Patients' pain immediately reported by students | Students | 4 (3–5) | 1 | / | / | 5 (4–5) | 4.07 (1.71–9.70) | / | / |
| D | D1) Distinguishing staff on training from HCWs | Residents | / | / | 3 (2–4) | 1 | 4 (2–5) | 2.13* (1.25–3.63) | 4 (2–4) | 1.20 (0.77–1.87) |
| | | Students | 4 (2–5) | 1 | / | / | 4 (4–5) | 5.43* (2.15–13.72) | 4 (3–4) | 0.85* (0.51–1.44) |
| | D2) Patients' trust | Residents | / | / | 4 (4–4) | 1 | 5 (5–5) | 10.20 (5.80–18.11) | 4 (3–5) | 1.10 (0.07–1.74) |
| | | Students | 3 (2–4) | 1 | / | / | 4 (4–5) | 15.23 (7.21–32.19) | 3 (2–3) | 0.77** (0.40–1.49) |
| | D3) Patients' satisfaction of staff on training assistance | Residents | / | / | / | / | 5 (5–5) | 1 | 4 (4–5) | 0.10 (0.05–0.18) |
| | | Students | / | / | / | / | 4 (3–5) | 1 | 3 (3–4) | 0.23* (0.11–0.48) |
| | D4) Adequate students flow within wards | Students | 2 (1–2) | 0.10 (0.05–0.19) | 2 (2–3) | 0.19 (0.10–0.36) | 4 (2–5) | 1 | 2 (1–3) | 0.09* (0.04–0.20) |

*: Generalized ordered logistic regression (1,2,3 scores vs 4,5 scores stratum), all strata in Supplementary Material S3 Appendix. Data set.

**: Logistic regression (1,2,3 scores vs 4,5 scores dichotomy).

/: Question not asked.

Students' ability to protect patient privacy results is similar to residents', but there are significant differences, with HCWs giving the lowest, nonetheless adequate, scores.

## Clinical risk management

Significantly different ratings of clinical risk management were given by stakeholders, with HCWs being more critical on residents concerning medical record management (Table 2, Question C1: 4[2–5]; OR[95%CI] = 0.09 [0.03–0.22]) and reporting errors in the performance of tasks (Table 2, Question C3: 4[3–5]; OR[95%CI] = 0.28 [0.13–0.60]).

Similar ratings were given by HCWs also about the students (Question C1: 4[3–5]; OR[95% CI] = 0.09 [0.48–017]; Question C3: 3[2–4]; OR[95%CI] = 0.15 [0.08–0.27]).

In addition, there is little disagreement between HCWs and residents about the appropriateness of obtaining informed consents (Table 2, Question C2) (H-L test on this question's model indicated poor fit, likely due to near total agreement among residents).

There is no agreement that residents, while carrying out their activities, properly wash their hands with water and/or chlorhexidine gel when performing their duties, which is even more evident among medical students (Table 2, Question C4).

Data show that when patients report feeling pain, students immediately call the nurse or the physician, which is confirmed by patients (5 [4–5]) (Table 2, Question C5).

## Patients' perception of staff on training

The interviewed categories partially agree that patients can correctly distinguish staff on training from physicians, yet patients seem to be more confident on this issue (about residents: 4 [2–5]; about students: 4 [4–5]) (Table 2, Question D1).

Patients' confidence in staff on training is significantly higher (about residents: 5[5–5] OR [95%CI] = 10.20 [5.80–18.11]; about students: 4[4–5] OR[95%CI] = 15.23 [7.21–32.19]) than students', residents' and HCWs' (Table 2, Question D2) perception. The level of overall patient satisfaction with the quality of the assistance offered by staff on training (about residents: 5[5–5]; about students: 4[3–5]) was higher than what perceived by HCWs (about residents: 4[4–5] OR[95%CI] = 0.10 [0.05–0.18]; about students: 3[3–4] OR[95%CI] = 0.23 [0.11–0.48]) (Table 2, Question D3). According to the patients, the flow of students in the wards is adequate, while other categories are more critical on this point (Table 2, Question D4).

All results are shown in Table 2 as median and IQR, OR and 95%CI, and are presented below for each thematic area.

## Discussion

The level of satisfaction of healthcare quality expressed by patients also derives from the impact that staff on training has on them [18], which can be positive or negative [19, 20]. In our case, patients' overall satisfaction with healthcare seems to be good, especially their judgment on staff on training is higher than HCWs', which is still quite good. Our data support the evidence that the presence of residents in hospitals improves the overall quality of healthcare [19, 21]; nevertheless, their role as peer educators for medical students [22, 23] must be considered as well. Even if staff on training do not believe that students add value to healthcare, data have shown that student-led activities are also valued by other stakeholders. Privacy protection is a key issue for healthcare professional's education and training, as stated by deontological codes and related legislation [24–26]; significant differences emerged in this specific area of quality from the analysis, in particular HCWs' perception of privacy protection by medical students is lower than what reported by patients' and students' self-assessment.

Moreover, in our case, patients confirmed being just partially able to distinguish staff on training from HCWs as previously mentioned [27], although this problem was not confirmed by Barksby [14]. Staff on training and HCWs also strongly confirmed patients' difficulties in distinguishing residents from physicians, but were more confident about patients' ability to correctly identify medical students. In our particular context, these difficulties could be due to a lack of marks or features that could help in distinguishing specific categories, suggesting a hypothetical solution in the form of chromatically different uniforms for students and staff.

Since there is a discrepancy between the staff on training self-perception and the assessment reported by the HCWs regarding medical record management, reporting errors and hand washing, audit and feedback procedures [28] and role modelling of HCWs should be enhanced, as well as training staff involvement in hospital activities for safety and quality improvement. This survey should be repeated in the future, including other university hospitals or healthcare facilities for comparative purposes, and the specific variables analysed in this first assessment should be further explored through further research.

## Limits and strengths of the study

The survey was conducted using questionnaires that were only partially comparable between stakeholders' categories, as some questions were only addressed to three out of four categories. Some difficulties in reaching nurses were found as they do not have an institutional email address; nevertheless, the only email invitation to participate turned out to be insufficient also as far as physicians and medical resident are concerned. Other stakeholders' characteristics not considered in this analysis (e.g. specific ward, patient's age and clinical condition, number of students per ward/unit), could have played a role as confounders [29] and should be taken into account in further research on this topic. Furthermore, this was a monocentric study and comparison with other studies is limited by the fact that they were conducted using different methods, at different times and with different levels of engagement. The non-respondent rate was quite high, 67.9%. Nonetheless, a strong point of this survey is the assessment of highlights or shortcomings mentioned by different categories of interviewed stakeholders within the academic hospital, as a helpful tool to identify such elements in their local context. Also, the survey is designed as a structured assessment, in which different stakeholders are asked about the same issues and compared with each other. The application of a Likert scale allows putting a set of opinions into quantifiable and objective evaluations. Potential improvement purposes include tailoring contact methods and timing to the subjects' characteristics, strengthening the engagement of managers and recipients' commitment, and conducting effective reminders.

## Conclusions

Within Udine Academic Hospital, staff on training appears to contribute positively to the overall healthcare quality and safety. Our findings suggest that patients are generally satisfied with the quality of healthcare provided by medical students. However, some shortcomings still persist, especially in relation to safety issues, which should be addressed primarily by academic healthcare institutions. Nevertheless, the study showed encouraging results that suggest improvement strategies for the near future that should be implemented.

## Supporting information

**S1 Table. Perception of staff on training by students, residents, patients and HCWs: Complete results from generalised ordered logistic regression.**
(DOCX)

**S1 Appendix. Questionnaires (English).**
(PDF)

**S2 Appendix. Questionnaires (Italian).**
(PDF)

**S3 Appendix. Data set.**
(XLS)

## Author Contributions

**Conceptualization:** Giuseppe Perri, Matteo d'Angelo, Laura Brunelli, Silvio Brusaferro.

**Data curation:** Cecilia Smaniotto, Massimo Del Pin, Luigi Castriotta.

**Formal analysis:** Massimo Del Pin, Luigi Castriotta.

**Methodology:** Giuseppe Perri, Matteo d'Angelo, Carla Londero, Laura Brunelli.

**Project administration:** Giuseppe Perri, Carla Londero, Laura Brunelli, Silvio Brusaferro.

**Visualization:** Carla Londero, Luigi Castriotta, Silvio Brusaferro.

**Writing – original draft:** Giuseppe Perri, Matteo d'Angelo, Cecilia Smaniotto, Massimo Del Pin, Edoardo Ruscio.

**Writing – review & editing:** Carla Londero, Laura Brunelli, Luigi Castriotta, Silvio Brusaferro.

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
