## [Decision Letter · Decision Letter 0]

18 Jun 2021

PONE-D-21-00656

What patients, healthcare workers, residents and students think about quality of care in an Italian academic hospital?

PLOS ONE

Dear Dr. Cecilia Smaniotto,

Thank you for submitting your manuscript to PLOS ONE. After careful consideration, we feel that it has merit but does not fully meet PLOS ONE’s publication criteria as it currently stands. Therefore, we invite you to submit a revised version of the manuscript that addresses the points raised during the review process.

We look forward to receiving your revised manuscript.

Kind regards,

Sharon Mary Brownie

Academic Editor

PLOS ONE

Journal Requirements:

Reviewers' comments:

Reviewer's Responses to Questions

**Comments to the Author**

1. Is the manuscript technically sound, and do the data support the conclusions?

Reviewer #1: Yes

Reviewer #2: No

2. Has the statistical analysis been performed appropriately and rigorously? 

Reviewer #1: No

Reviewer #2: No

3. Have the authors made all data underlying the findings in their manuscript fully available?

Reviewer #1: No

Reviewer #2: No

4. Is the manuscript presented in an intelligible fashion and written in standard English?

Reviewer #1: Yes

Reviewer #2: Yes

5. Review Comments to the Author

Reviewer #1: Thanks for the authors for the focus on quality of care, which I think is a bit neglected research area. The study definitely has a good merit.

Abstract

1. I didn't see and understand what the study design is

2. You mentioned the total questionnaires were 579/1,813, but, if you sum up the total, which is 165/772 HCWs, 111/355 residents, 121/389 students and 200/347 patients it gives 597/1863, can you explain that

Background

1. The background section is a little too much, although it is informative. I would subject narrowing it to less than 2 pages

Data analysis

1. You mentioned Ordered logistic regression were performed. Given that your outcome variable is ordinal it is the appropriate method. However, you didn't write about the assumptions that needs to be satisfied to proceed with ordered logit. The parallel lines assumption should be tested. If it fails, you should conduct generalized ordered logistic regression. This is quiet critical to make sure the results are valid.

Reviewer #2: A paper entitled “What patients, healthcare workers, residents, and students think about the quality of care in an Italian academic hospital?” is seeking to view the quality of care provided in X hospital by involving different stakeholders. Such articles have a good input towards the improvement of health care delivery. Here are some issues I encountered while reviewing the paper.

1. The title of the study is quite different from the body (main content).

2. Since the title is about perception, it is better if it had a qualitative method of data collection coupled with the qualitative one.

3. Line number 29, what is the need for the word “respectively”? What series was it represent?

4. Line number 29, the sum of subjects the study offered was 1863, not 1813 and also response collected was 597, not 579, according to your pieces of data.

5. Line number 113, the Main objective is totally different from the title of the study. The title states about the perception of overall quality of health care, whereas the main objective stated about the perception of different stakeholders towards the attendance of staff on training. The title seems a bit wide, it is better if modification is made based on the objective.

6. The main objective and the specific objective are not addressing well in the study.

7. Line number 141, inclusion criteria state that participants willing to participate in the survey were included whereas one criterion put in the exclusion criteria was a refusal to participate. Since study subjects refuse to can’t fulfill inclusion criteria consequently it cannot be an exclusion criterion.

8. The conclusion is not in accordance with the findings.

9. Line number 163, expected response rate 50%, if the non-respondent rate becomes as high as 50% it will not be representative of the population and also considering those non-respondent characteristics is also important for example if they are homogeneous in character it is difficult to exclude since they have something in common.

10. Line number 179-183, summation error noticed in the total study offered.

11. Line number 179, 1,813 subjects non-respondent rate were not mentioned how much was it?

12. The use of two words/phrases, “staff on training” vs “medical students and residents”, interchangeable are creating a little bit of confusion throughout the paper, better choose one.

Overall title, objective, and body of the manuscript lack coherence.

6. PLOS authors have the option to publish the peer review history of their article (what does this mean?). If published, this will include your full peer review and any attached files.

Reviewer #1: No

Reviewer #2: No

---

## [Author Response · Author response to Decision Letter 0]

30 Jul 2021

Udine, July 29th, 2021

To the kind attention of dr. Sharon Mary Brownie (Academic Editor, Plos One),

Please find here enclosed the revised version of the manuscript entitled “Does staff on training have an impact on quality of care? An assessment from different stakeholders in an Italian academic hospital, 2019.” by Perri G, d’Angelo M, Smaniotto C, Del Pin M, Ruscio E, Londero C, Brunelli L, Castriotta L, Brusaferro S. 

The manuscript now includes the alterations required by Reviewer 1 and Reviewer 2 to better present the research. More specifically, the manuscript has been modified as follows:

a) Reviewer 1:

1) The study design is now more specifically described in the Methods section.

2) The number of collected questionnaires and interviewed subjects is now correct.

3) Data analysis was completed with the explanation and a new reference for the assumptions to use generalised ordered logistic regression.

b) Reviewer 2:

1) The background section has been shortened.

2) A quantitative method of data collection coupled with the qualitative one could be used in the next edition of the study. For the study carried out in 2019, the method used in the pilot study of 2017 was kept.

3) Line 29: “Respectively” has been removed..

4) The number of collected questionnaires and interviewed subjects is now correct.

5) The title has been modified from “What patients, healthcare workers, residents and students think about quality of care in an Italian academic hospital?” to “Does staff on training have an impact on quality of care? An assessment from different stakeholders in an Italian academic hospital, 2019.” to better match the main objective.

6) The explanation of main objective and secondary objectives has been modified in order to have a better concordance with both title and main text.

7) Refusing to participate to the study has been removed from the exclusion criteria as being willing to participate is already an inclusion criterion.

8) Conclusions have been modified in order to have a better concordance with results and discussion.

9) Expected response rate for patients was lower than expected response rate for the other stakeholder categories as inpatiens could be unable to partecipate albeit willing, due to their physical or psychological conditions at the moment of the questionnaire collection. Assuming a 70% expected responde rate for inpatients, the minimum number of needed questionnaires would have been 66.

10) Line 175-179: the number of collected questionnaires and interviewed subjects is now correct.

11) Line 269: non-respondent rate is now mentioned.

12) When referring to both categories together, “staff on training” is now used to describe “medical students and residents”.

The minimum anonymised data set to replicate the study is now available as a Supporting Information file. Four Supporting Information files are now available, including two appendixes for questionnaires (Italian and English version). A new table presents the complete results from generalised ordered logistic regression

All authors have read and approved the manuscript. The paper has not been published, and is not under review elsewhere. There are no ethical problems or conflicts of interest.

We thank you in advance for your kind consideration.

With kindest regards

Sincerely Yours,

Cecilia Smaniotto, MD

Corresponding author

Department of Medicine, University of Udine. 

Address: Via Colugna 50, 33100 Udine, Italy. 

Phone +390432554767. Email: smaniotto.cecilia@spes.uniud.it

---

## [Decision Letter · Decision Letter 1]

8 Sep 2021

PONE-D-21-00656R1Does staff on training have an impact on quality of care? An assessment from different stakeholders in an Italian academic hospital, 2019.PLOS ONE

Dear Dr. Cecilia Smaniotto,

Thank you for submitting your manuscript to PLOS ONE. After careful consideration, we feel that it has merit but does not fully meet PLOS ONE’s publication criteria as it currently stands. Therefore, we invite you to submit a revised version of the manuscript that addresses the points raised during the review process. Please submit your revised manuscript by 8 October. If you will need more time than this to complete your revisions, please reply to this message or contact the journal office at plosone@plos.org. Please include the following items when submitting your revised manuscript:A rebuttal letter that responds to each point raised by the academic editor and reviewer(s). You should upload this letter as a separate file labeled 'Response to Reviewers'.A marked-up copy of your manuscript that highlights changes made to the original version. You should upload this as a separate file labeled 'Revised Manuscript with Track Changes'.An unmarked version of your revised paper without tracked changes. You should upload this as a separate file labeled 'Manuscript'.If applicable, we recommend that you deposit your laboratory protocols in protocols.io to enhance the reproducibility of your results. Protocols.io assigns your protocol its own identifier (DOI) so that it can be cited independently in the future. For instructions see: https://journals.plos.org/plosone/s/submission-guidelines#loc-laboratory-protocols. Additionally, PLOS ONE offers an option for publishing peer-reviewed Lab Protocol articles, which describe protocols hosted on protocols.io. Read more information on sharing protocols at https://plos.org/protocols?utm_medium=editorial-email&utm_source=authorletters&utm_campaign=protocols.

We look forward to receiving your revised manuscript.

Kind regards,

Sharon Mary Brownie

Academic Editor

PLOS ONE

Journal Requirements:

Reviewers' comments:

Reviewer's Responses to Questions

**Comments to the Author**

1. If the authors have adequately addressed your comments raised in a previous round of review and you feel that this manuscript is now acceptable for publication, you may indicate that here to bypass the “Comments to the Author” section, enter your conflict of interest statement in the “Confidential to Editor” section, and submit your "Accept" recommendation.

Reviewer #2: All comments have been addressed

Reviewer #3: All comments have been addressed

2. Is the manuscript technically sound, and do the data support the conclusions?

Reviewer #2: Yes

Reviewer #3: Yes

3. Has the statistical analysis been performed appropriately and rigorously? 

Reviewer #2: Yes

Reviewer #3: Yes

4. Have the authors made all data underlying the findings in their manuscript fully available?

Reviewer #2: Yes

Reviewer #3: Yes

5. Is the manuscript presented in an intelligible fashion and written in standard English?

Reviewer #2: Yes

Reviewer #3: Yes

6. Review Comments to the Author

Reviewer #2: (No Response)

Reviewer #3: The authors have sufficiently addressed all the reviewers’ comments. However, the manuscript needs some minor revisions. First, in the result section, the authors should clearly highlight which results are referred to from the table 2. It is currently difficult to follow through the results on the table against the results in text. The authors should consider highlight the odds ratio / medians like in the abstract and refer to the table. Second, some typographical and grammatical errors should be corrected including the presentation on “n” in lines 185-193. Third, subtitles should be used in the method section (line 109-148) for better readability and clarity. Fourth, the authors should consider revising the title from "Does staff of training have an impact on quality of care?" to "Do medical students and residents impact the quality of patient care?". Similarly, the reference to "staff of training" should be revised to "medical students and residents" for clarity and readability.

7. PLOS authors have the option to publish the peer review history of their article (what does this mean?). If published, this will include your full peer review and any attached files.

Reviewer #2: No

Reviewer #3: No

---

## [Author Response · Author response to Decision Letter 1]

30 Sep 2021

Udine, September 28th, 2021

To the kind attention of dr. Sharon Mary Brownie (Academic Editor, Plos One),

Please find here enclosed the revised version of the manuscript entitled “Does staff on training have an impact on quality of care? An assessment from different stakeholders in an Italian academic hospital, 2019.” by Perri G, d’Angelo M, Smaniotto C, Del Pin M, Ruscio E, Londero C, Brunelli L, Castriotta L, Brusaferro S. It is now entitled “Do medical students and residents impact the quality of patient care? An assessment from different stakeholders in an Italian academic hospital, 2019.”

The manuscript now includes the alterations required by Reviewer 3 to better present the research. The manuscript has been modified as follows:

1) The results referred to Table 2 are now more clearly highlighted in the Results section. 

2) The errors in line 185-193 have been corrected.

3) More subtitles have been added in the methods section (Design of the study and previous research; Design of the questionnaire and testing phase; Inclusion and exclusion criteria; Data collection).

4) The title has been revised as suggested. Nonetheless, the previous reference to “medical students and residents” has been kept as this was the suggestion of Reviewer 2 in Revision 1, and it was applied throughout the text. The authors can change again the reference back to “staff on training”, however, if this would improve clarity and readability of the article.

All authors have read and approved the manuscript. The paper has not been published, and is not under review elsewhere. There are no ethical problems or conflicts of interest.

We thank you in advance for your kind consideration.

With kindest regards

Sincerely Yours,

Cecilia Smaniotto, MD

Corresponding author

Department of Medicine, University of Udine. 

Address: Via Colugna 50, 33100 Udine, Italy. 

Phone +390432554767. Email: smaniotto.cecilia@spes.uniud.it

---

## [Editor Report · Decision Letter 2]

4 Oct 2021

Do medical students and residents impact the quality of patient care? An assessment from different stakeholders in an Italian academic hospital, 2019.

PONE-D-21-00656R2

Dear Dr. Cecilia Smaniotto,

We’re pleased to inform you that your manuscript has been judged scientifically suitable for publication and will be formally accepted for publication once it meets all outstanding technical requirements.

Kind regards,

Sharon Mary Brownie

Academic Editor

PLOS ONE

Editor Comments 
---

## [Editor Report · Acceptance letter]

7 Oct 2021

PONE-D-21-00656R2 

Do medical students and residents impact the quality of patient care? An assessment from different stakeholders in an Italian academic hospital, 2019. 

Dear Dr. Smaniotto:

I'm pleased to inform you that your manuscript has been deemed suitable for publication in PLOS ONE. Congratulations! Your manuscript is now with our production department. 

Kind regards, 

on behalf of

Professor Sharon Mary Brownie 

Academic Editor

PLOS ONE